# Direct evidence of spatial stability of Bose-Einstein condensate of magnons

I. V. Borisenko [1,2 ✉], B. Divinskiy[1], V. E. Demidov[1], G. Li[3], T. Nattermann[4], V. L. Pokrovsky[3,5] & S. O. Demokritov[1]

Bose-Einstein condensation of magnons is one of few macroscopic quantum phenomena observed at room temperature. Since its discovery, it became an object of intense research, which led to the observation of many exciting phenomena such as quantized vortices, second sound, and Bogolyubov waves. However, it remained unclear what physical mechanisms can be responsible for the spatial stability of the magnon condensate. Indeed, since magnons are believed to exhibit attractive interaction, it is generally expected that the condensate is unstable with respect to the real-space collapse, contrarily to experimental findings. Here, we provide direct experimental evidence that magnons in a condensate exhibit repulsive inter- action resulting in the condensate stabilization and propose a mechanism, which is responsible for this interaction. Our experimental conclusions are additionally supported by the theoretical model based on the Gross-Pitaevskii equation. Our findings solve a long- standing problem, providing a new insight into the physics of magnon Bose-Einstein condensates.

[1] Institute for Applied Physics and Center for Nanotechnology, University of Muenster, 48149 Muenster, Germany. [2] Kotel'nikov Institute of Radio Engineering and Electronics, Russian Academy of Sciences, Moscow, Russia 125009. [3] Department of Physics and Astronomy, Texas A&M University, College Station, TX 77843-4242, USA. [4] Institute of Theoretical Physics, University of Cologne, Zülpicher Strasse 77, 50937 Köln, Germany. [5] Landau Institute of Theoretical Physics, Russian Academy of Sciences, Chernogolovka, Moscow Region, Russian Federation 142432. ✉email: boriseni@uni-muenster.de

The discovery of the room-temperature magnon Bose–Einstein condensation (BEC) in yttrium-iron garnet (YIG) films driven by parametric pumping[1] has spurred intense experimental and theoretical studies of this phenomenon[2–12] and opened a new field in modern magnetism—room temperature quantum magnonics. The formation of magnon BEC driven by the parametric pumping has been experimentally confirmed by the observation of the spontaneous narrowing of the population function in the energy[1,3,9] and the phase[4] space. Moreover, the spontaneous coherence of this state has been proven by the observation of interference between two condensates in the real space, as well as by the observation of formation of quantized vortices[10].

In spite of all these experimental observations, from the theoretical point of view, the possibility of magnon BEC is still questioned because of the issues associated with the condensate stability[13]. It is generally believed that the formation of a stable Bose–Einstein condensate is possible only if the particle scattering length $a$ proportional to the inter-particle interaction coefficient $g$ is positive (the so-called repulsive particle interaction)[14,15]. Otherwise, the condensate is expected to collapse[16,17]. However, the established theories of magnon–magnon interactions in unconfined ferromagnets[18–21] with uniaxial anisotropy predict an attractive interaction between magnons, which was confirmed experimentally by numerous studies of magnetic solitons in films with weak in-plane[22–24] and strong out-of-plane[25,26] anisotropy. The problem of interactions becomes particularly complex for magnons existing in in-plane magnetized films in the vicinity of the lowest-energy state, where the BEC is formed, since conventional interaction mechanisms originating from the dipolar shape anisotropy weaken strongly in this spectral part and other mechanisms, such as the interaction between condensed and non-condensed magnons[11,27] can become dominant. Up to now, the clear understanding of these issues was missing resulting in a deep contradiction between the experiments demonstrating the possibility of stable magnon BEC and the theory predicting its collapse.

Here, we provide a direct experimental evidence of stability of magnon BEC and of repulsive character of magnon–magnon interaction in the vicinity of the lowest-energy spectral state. By using a confining potential, we create a strong local disturbance of the condensate density and study the spatio-temporal dynamics of BEC. Our experimental data clearly show that, in spite of the artificially created strong local increase of the condensate density expected to stimulate the collapse process, no collapse occurs. On the contrary, the observed behaviors clearly indicate that magnons forming the condensate experience repulsive interaction, which counteracts an accumulation of magnons in a particular spatial location. These conclusions are well supported by calculations using a model based on the Gross–Pitaevskii equation. We also propose a mechanism, which is likely responsible for the observed repulsive interaction. We show that the effective magnon repulsion can be associated with the influence of additional dipolar magnetic fields appearing in response to any local increase of the condensate density. This interpretation is well supported by the quantitative analysis of the experimental data. Our findings resolve the long-standing question of stability of magnon condensates.

## Results

**Studied system and experimental approach.** We study the spatio-temporal dynamics of room-temperature magnon BEC in a YIG film using the experimental setup similar to that used in[11,28]. The schematic of the experiment is illustrated in Fig. 1a, b. The condensate in YIG film with the thickness of 5.1 μm and the lateral dimensions of $2 \times 2$ mm is created by a microwave parametric pumping using a dielectric resonator with the resonant frequency of $f_p = 9.055$ GHz. The pumping injects primary magnons at the frequency of $f_p/2$, which thermalize and create BEC in the lowest-energy spectral state[1,2]. The frequency of BEC is determined by the magnitude of the static magnetic field $H_0$, which was varied in the range 0.5–1.5 kOe. Below we discuss the representative data obtained at $H_0 = 0.64$ kOe corresponding to the BEC frequency of 1.9 GHz and the wavelength of about 1 μm. To study the spatial stability of the condensate, we use a confining potential, which is created by using an additional spatially inhomogeneous magnetic field $\Delta H$ induced by a dc electric

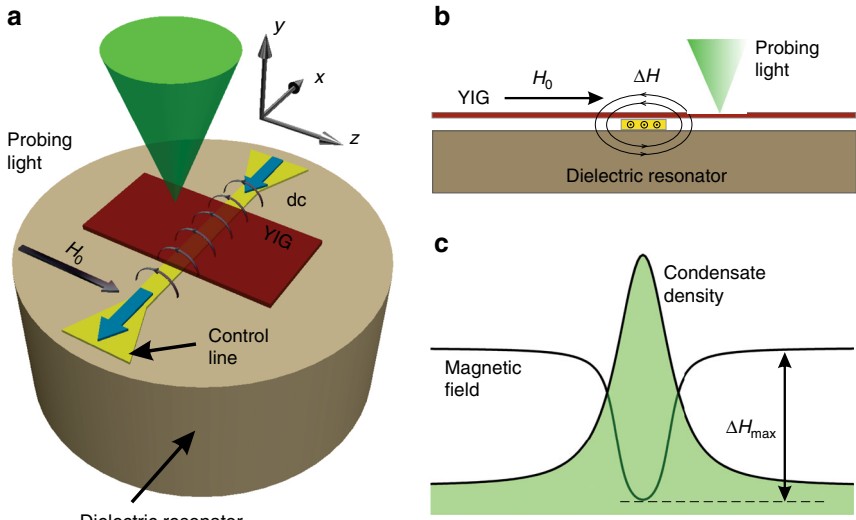

**Fig. 1 Schematic of the experiment. a** General view of the experimental system. Dielectric resonator creates a microwave-frequency magnetic field, which parametrically excites primary magnons in the YIG film. After thermalization, the magnons form BEC in the lowest-energy spectral state. dc electric current in the control line placed between the resonator and the YIG film, produces a non-uniform magnetic field, which adds to the uniform static field $H_0$. The local density of condensed magnons is recorded by BLS with the probing laser light focused onto the surface of the YIG film. **b** Cross-section of the experimental system illustrating the field created by the control line. **c** Spatial distribution of the horizontal component of the total magnetic field $H_0 + \Delta H$ and the corresponding spatial profile of the condensate density caused by the inhomogeneity of the field.

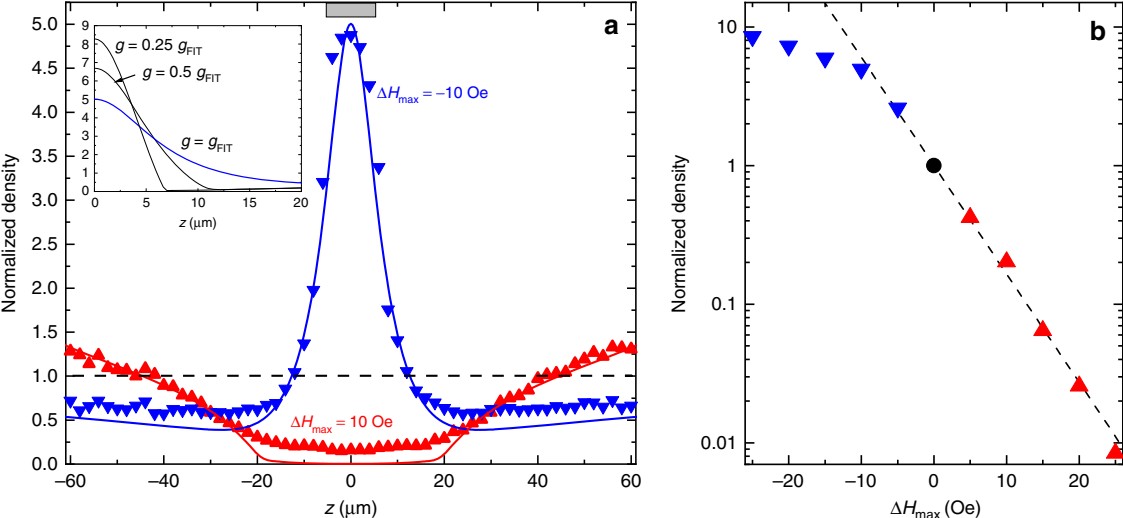

**Fig. 2 Spatial redistribution of the condensate density caused by a potential well and a hill. a** Representative profiles of the condensate density recorded by BLS for the case of a potential well (blue symbols) and a potential hill (red symbols) with the depth/height of 10 Oe. The shown profiles are normalized by the value of the density measured in the absence of the potential. The gray rectangle in the upper part marks the position of the control line creating the inhomogeneous field potential. Solid curves show the results obtained from numerical solution of Eq. (8). Inset: Spatial profiles of the condensate density calculated for different magnitudes of the coefficient $g$ describing the nonlinear magnon–magnon interaction with all other parameters fixed. The curve labeled $g = g_{FIT}$ is the same, as that in the main figure. **b** Normalized condensate density at the center of the hill/well versus $\Delta H_{max}$ in the log-linear scale (the color coding is the same as in **a**). The dashed line is the exponential fit of the data for $\Delta H_{max} > 0$.

current flowing in a control strip line with the width of 10 μm (Fig. 1c).

We study the condensate density $n$ and, in particular, its spatial and temporal variation using micro-focus Brillouin light scattering (BLS) technique[29,30]. In stationary-regime experiments, both the pumping and the inhomogeneous field are applied continuously. Additionally, to investigate the temporal dynamics of the condensate, we perform experiments, where the pumping is applied in the form of pulses with the duration of 1 μs and the repetition period of 10 μs and the BLS signal was recorded as a function of the time delay with respect to falling edge of the pumping pulse.

**BEC in static inhomogeneous potentials**. Figure 2a shows representative spatial profiles of the condensate density recorded by BLS for the case of a potential well (blue symbols) and a potential hill (red symbols) with the depth/height of 10 Oe. The shown profiles are normalized by the value of the density measured in the absence of the potential. The recorded BLS intensity is proportional to the condensate density. However, since the Au control line causes additional reflections of the light, the factor, connecting these two values is larger in the area under the line, than that in the rest of the film. To avoid the influence of this spatially varying factor, we show and analyze the normalized profiles, where the effects of the non-uniform reflections are canceled.

A gray rectangle in the upper part of Fig. 2a marks the position of the control line creating the inhomogeneous field potential. The data of Fig. 2a show that, in the case of a potential well ($\Delta H_{max} = -10$ Oe), the profile $n(z)$ exhibits an intense peak of the condensate density centered at the middle of the control line and a noticeable reduction of the density outside the potential well. We associate these observations with a directional transport of condensed magnons in the potential gradient and their accumulation in the middle of the well, where the potential energy of magnons minimizes. This interpretation is also consistent with the observed reduction of the magnon density in the regions surrounding the well. The same mechanisms

govern the distribution of the condensate density in the case of a potential hill ($H_{max} = +10$ Oe): the condensed magnons tend to leave the area of the increased field, which results in the reduced condensate density in the middle of the potential hill and its increase in the surrounding regions. The solid curves in Fig. 2a show the density distributions calculated using the theoretical model described below. Although the agreement between the theory and the experiment is not perfect, Fig. 2a demonstrates that the model describes well the gross features of the phenomenon. Figure 2b shows the dependence of the normalized condensate density as a function of the height of the potential hill ($\Delta H_{max} > 0$) and the depth of the potential well ($\Delta H_{max} < 0$). The data of Fig. 2b indicate that the condensate density decreases exponentially with the increase of the height of the potential hill by more than two orders of magnitude. In contrast, for a potential well, the density of the accumulated magnons increases with the increase of the depth of the well at a much smaller rate.

We emphasize that these behaviors are inconsistent with the assumption of the attractive magnon–magnon interaction. Indeed, if the interaction were attractive, one would expect a diverging increase of the magnon density with the increasing magnitude of $\Delta H_{max} < 0$ followed by a condensate collapse. Therefore, the data of Fig. 2a clearly indicate that this assumption is not correct. Moreover, as discussed in details below, the observed behaviors require the existence of significant repulsive interaction.

**Spatio-temporal dynamics of the condensate**. The data presented in Fig. 2 describe the stationary state of the condensate, which is determined by a complex balance between three different processes: (i) thermalization of injected magnons and their condensation, (ii) decay of the condensate density due to spin–lattice relaxation, and (iii) lateral transport of magnons due to the potential gradient and magnon–magnon interactions. To get a better insight into intrinsic properties of the condensate, we analyze the temporal dynamics of the condensate following turning the microwave pumping field off. Taking into account

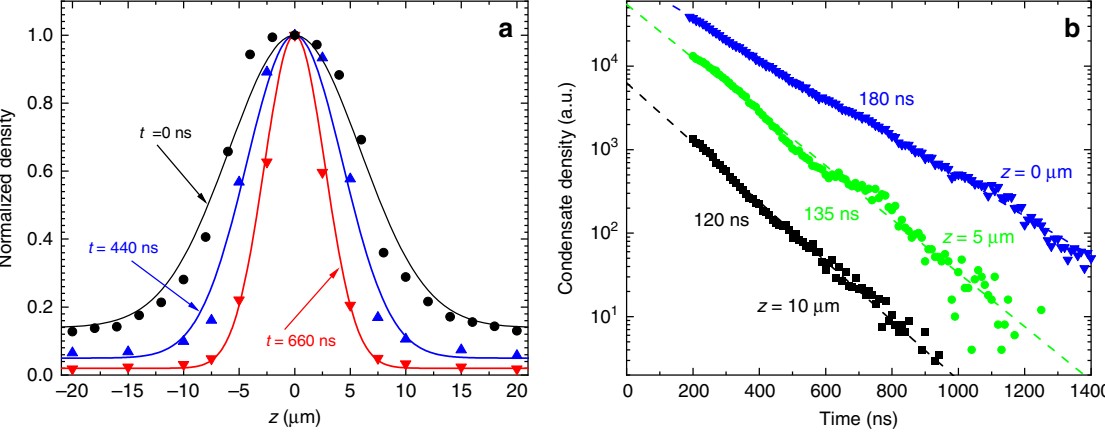

**Fig. 3 Time evolution of the condensate density in a potential well after turning the microwave pumping off. a** Normalized spatial profiles of the condensate density in a potential well with $\Delta H_{max} = -10$ Oe recorded at different delays after the microwave pumping is turned off at $t = 0$. Solid lines are guides for the eye. **b** Temporal dependence of the condensate density at different spatial positions. Dashed lines show the exponential fit of the experimental data with the corresponding effective decay times as indicated.

that the typical magnon thermalization time is below 100 ns[2], at delays of more than 300 ns after the end of the pumping pulse, the process (i) does not influence the density distribution anymore. Since, with a good accuracy, the process (ii) can be considered as independent of the condensate density and of the non-uniform potential landscape, the free dynamics of the condensate density is mostly governed by the process (iii).

The free dynamics of the condensate spatial distribution is characterized in Fig. 3. Figure 3a shows the normalized profiles of the condensate density in a potential well at different delays after the pumping is turned off at $t = 0$ and the condensed magnons start to relax into the sample lattice. If the interaction between magnons were attractive, the overall reduction of the condensate density would result in a spatial broadening of the density peak. Indeed, the attraction between magnons are more efficient in a dense gas resulting in their stronger spatial concentration. In contrast, the data of Fig. 3a show opposite behaviors: after the pumping is turned off, the density distribution $n(z)$ starts to narrow. We associate this finding with the effects of the repulsive interaction between magnons. Such interaction should result in a strong initial broadening of the profile at large densities achieved during the pumping pulse. However, after the pumping is turned off, the density of magnons in BEC reduces due to their relaxation and the broadening cased by the repulsion becomes less efficient. Accordingly, at small densities, the spatial width of the profile is predominantly determined by the lateral transport of condensate magnons in the potential gradient causing their stronger accumulation at the bottom of the potential well.

This scenario is further supported by the data of Fig. 3b, which illustrates the temporal decay of the condensate density at different spatial locations: although, in all points, the decay is exponential, it is slower at the center of the potential well ($z = 0$) and becomes noticeably faster with the increase of the distance from the center. Taking into account that the relaxation to the lattice is position-independent, the difference in decay rates can only be associated with a redistribution of condensed magnons in the real space. With decreasing total density, the magnons that were previously pushed away from the center of the well by the repulsive interaction, move back to the center under the influence of the potential gradients, which results in an effective reduction of the decay rate at this spatial position.

**Theoretical description.** Before turning to the theoretical model describing the experimental findings, let us address the issue of

the condensate coherence. For a fully coherent condensate placed in a well, one expects formation of quantized energy levels. However, the estimations based on the parameters of the well and the magnon dispersion spectrum show that the frequencies of the neighboring levels differ by 1–2 MHz, which is close to the measured linewidth of the condensate[9]. In agreement with this estimation, the measured frequency of the condensate (Fig. 4a) demonstrates a continuous variation indicating a strong hybridization of quantized levels. Thus, for the realistic theoretical description of the condensate, it is necessary to take into account dissipative processes in the magnon gas.

Since in the experiment the magnetic field varies in the $z$-direction only, we consider the condensate wave function in the one-dimensional form:

$$\Psi(z) = \psi(z)e^{ik_0z} = \sqrt{n(z)}e^{i\varphi(z)}e^{ik_0z}. \qquad (1)$$

Here $\psi(z)$ is the condensate envelope function with $\varphi$ being its phase, $n$ is the density of magnons in the condensate, and $k_0 = 5 \times 10^4$ cm$^{-1}$ is the wavevector of the condensate ground state, corresponding to the minimum magnon frequency. Correspondingly, the one-dimensional Gross-Pitaevskii equation, describing the condensate with a constant total number of magnons:

$$i\hbar\frac{\partial\psi}{\partial t} = \left(-\frac{\hbar^2}{2m}\frac{\partial^2}{\partial z^2} + U(z) + g|\psi|^2 - \mu_C\right)\psi. \qquad (2)$$

Here $m$ is the positive effective mass of magnons defined by the curvature of the magnon dispersion curve close to the minimum frequency, $U(z)$ is a potential energy of the condensate mostly determined by the applied magnetic field $U(z) \approx 2\mu_B H(z)$, $\mu_C$ is the chemical potential, and $g$ is the magnitude of magnon–magnon interaction. Note here that the expression $U(z) + g|\psi|^2$ defines an effective potential energy of the condensate per magnon. Accordingly, $g > 0$ corresponds to the magnon repulsion. The flux of the probability density in the $z$-direction $J = -\frac{i\hbar}{2m}\left(\psi^*\frac{\partial\psi}{\partial z} - \psi\frac{\partial\psi^*}{\partial z}\right)$ can be written as follows:

$$J = nv, \qquad (3)$$

where $v = \frac{\hbar}{m}\frac{\partial\varphi}{\partial z}$ is velocity of the condensate. Substituting Eqs. (1, 3) into Eq. (2) and separating the real and the imaginary parts, on obtains for the imaginary part:

$$\frac{\partial n}{\partial t} = -\frac{\partial(nv)}{\partial z}. \qquad (4)$$

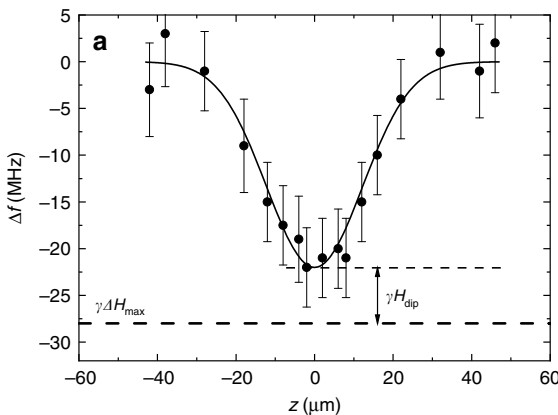
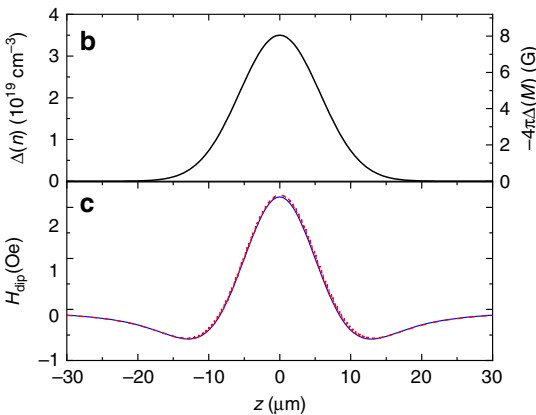

**Fig. 4 Spatial distributions of the condensate frequency and of the induced dipolar field. a** Spatial profile of the frequency shift of the condensate caused by the presence of the potential well ($\Delta H_{max} = -10$ Oe). Symbols—experimental data obtained using BLS with an ultimate frequency resolution. Solid curve—fit of the experimental data by a Gaussian peak. Horizontal dashed line marks the expected maximum frequency shift, $-28$ MHz caused by $\Delta H = -10$ Oe. Error bars are standard deviation. **b** Non-uniform distribution $\Delta n(z)$ used for calculations of the dipolar field $H_{dip}$. **c** Resulting distribution of the dipolar field $H_{dip}$ calculated analytically (solid curve) and numerically (dashed curve).

or

$$\frac{\partial n}{\partial t} = -\frac{\partial(nv)}{\partial z} - \frac{n - n_0}{\tau}, \qquad (5)$$

where the last term in Eq. (5) is added to take into account processes changing the total number of magnons in the condensate, which are: (i) continuous condensation of incoherent magnons into the condensate and (ii) relaxation of the condensate due to the spin–lattice coupling. Here the relaxation time $\tau$ is the time, during which the density of a free condensate decreases by a factor of $e$.

Correspondingly, by differentiating of the real part, one obtains, neglecting the highest-order derivative because of its small contribution for the slowly varying potential:

$$m\frac{dv}{dt} = -\frac{\partial}{\partial z}(U + gn) \qquad (6)$$

Similar to Eq. (5), one should modify Eq. (6) in order to take into account dissipation. As usual for dissipative processes, one can write the connection between the effective potential $U+gn$ and drift velocity in the form:

$$v = -\eta\frac{\partial}{\partial z}(U + gn), \qquad (7)$$

where $\eta$ is the mobility of the condensate, defined by the momentum relaxation processes. It is important to notice that, since the mobility of non-condensed magnons strongly differs from that of the condensate, the corresponding velocity induced by the potential gradients are different. Thus, the interaction of the condensate with non-condensed magnons represents an intrinsic process of the momentum relaxation of the condensate.

After substitution of Eq. (7) into Eq. (5), one obtains for the stationary case $\frac{\partial n}{\partial t} = 0$ the final non-linear second-order differential equation for the condensate density $n$:

$$\eta\frac{\partial}{\partial z}\left[n\frac{\partial(U + gn)}{\partial z}\right] - \frac{n - n_0}{\tau} = 0, \qquad (8)$$

which can be solved numerically with $g$ and $\eta$ as fit parameters. Note that the relaxation time $\tau = 130$ ns, was independently measured for homogeneous condensate by analyzing the temporal dependence of the condensate density after abrupt switching off the pumping.

## Discussion

The distribution $n(z)$ obtained from the numerical solution of Eq. (8) in the case of the well for $g = 7 \times 10^{-39}$ erg cm$^3$ and $\eta = 4.5 \times 10^{21} s/g$ is shown in Fig. 2a (solid blue line). As seen from the figure, the simple nonlinear model presented above describes the experimental data very well. Note that the theory reproduces not only an expected maximum of the condensate density at the center of the potential well, but also a counter-intuitive reduction of the density in the surroundings. We emphasize, that the reasonable agreement between the calculations and the experiment can only be achieved for $g > 0$ corresponding to the repulsive interaction between magnons. The inset in Fig. 2a illustrates the effects of the repulsive nonlinearity on the density distribution: reduction of $g$ inevitably causes a growth of the $n(z)$ peak. Moreover, for $g = 0$, no numerical solution can be found: the condensate collapses, in agreement with the general picture of the phenomenon. This fact, together with the experimentally observed growth of the peak height accompanying the overall reduction of the condensate density (Fig. 3a) provides a strong evidence for the existence of a repulsive magnon–magnon interaction and its decisive role for the condensate stability.

Our model also describes well the density profile for the case of the potential hill. The red curve in Fig. 2a was obtained for the same value of $g = 7 \times 10^{-39}$ erg cm$^3$ and $\eta = 1.8 \times 10^{22} s/g$, which is larger than that for the case of the potential well due to the smaller density of non-condensed magnons, which mainly determine the momentum relaxation of the condensate.

To understand the contradiction between our results and those of earlier theoretical works[7,13,19,31], one should take into account that, in the previous works, the analysis was performed for the case of a uniform spatial distribution of the condensate density. However, any instability of the condensate causing its collapse in the real space, leads to a non-uniform distribution of $n$, which, as will be shown below, can result in an effective repulsive magnon–magnon interaction. To illustrate this, let us consider a non-uniform distribution $\Delta n(z)$ leading to a spatially dependent reduction of the static magnetization of the film, $\Delta M = -2\mu_B \Delta n(z)$ (Fig. 4b). In a film with a finite thickness $d$, such a non-uniformity of the magnetization causes an appearance of an additional static "demagnetizing" dipolar field $H_{dip}$, which modifies the energy per magnon by $2\mu_B H_{dip}$. The profiles of $H_{dip}(z)$ calculated both analytically and by using numerical micromagnetic code Mumax3 are shown in Fig. 4c. Since the field

$H_{dip}$ is mostly parallel to the static magnetization, it increases the potential energy per magnon. Thus, any local increase of the magnon density causes a higher energy per magnon, which corresponds to magnon–magnon repulsion or positive $g$, in agreement with our experimental data. It might be surprising that a small dipole field $H_{dip} \ll H_0$ changes the sign of the scattering amplitude. We would like to emphasize that the frequency of the lowest state which is very close to the Larmor frequency $f_0 = \gamma H_0$ depends on $H_0$, but is independent of the magnon density. Since magnon scattering amplitude is determined by the dependence of the energy of magnons on their density, for magnons with the frequency $f_0$, the scattering amplitude is zero at any value of $H_0$. In contrast, a non-uniform spatial distribution of magnons results in appearance of the dipolar field, which linearly depends on the magnon density resulting in a density-dependent magnon energy. In other words, the effect of the non-uniformity-induced demagnetizing dipolar field on the magnon scattering amplitude is superior to that of the static uniform magnetic field $H_0$.

The increase of the energy associated with $H_{dip}$ can be measured directly, as illustrated in Fig. 4a. The horizontal dashed line marks the expected frequency shift of the condensate due to the reduced static field at the center of the potential well $\Delta f(z = 0) = \gamma \Delta H_{max} = -28$ MHz ($\gamma = 2.8$ MHz/Oe is the gyromagnetic ratio) corresponding to $\Delta H_{max} = -10$ Oe. The solid line shows a fit of the condensate frequency shift obtained from the experiment. One sees, that the measured frequency shift $-22 \pm 1$ MHz is smaller than the expected one, likely due to the influence of the additional dipolar field. From this difference, we calculate $\gamma H_{dip} = 6 \pm 1$ MHz and $H_{dip} = 2.1 \pm 0.3$ Oe. Although this field is much smaller than $H_0$, contrary to $H_0$, it contributes to the magnon–magnon interaction, since its value depends on the magnon density.

Based on the found value of $H_{dip}$ and on the profile $n(z)$ shown in Fig. 2a, we estimate the density of magnons in the unperturbed condensate $n_0 \approx 8 \times 10^{18}$ cm$^{-3}$. This value agrees well with that predicted by the BEC theory[1], which proves the validity of the proposed mechanism. Note here that the obtained value is by about three orders of magnitude smaller than the maximum possible density of magnons corresponding to a 100%-reduction of the magnetization, $10^{22}$ cm$^{-3}$.

One should admit that, as seen in Fig. 4b, the exact connection between $\Delta n(z)$ and $H_{dip}(z)$ is non-local due to the long-range character of the dipole interaction. A rigorous theory taking into account this fact, results in an integro-differential equation for $n(z)$ instead of the relatively simple Eq. (8) derived by us and used for the analysis of the experimental data. Moreover, a rigorous hydrodynamic theory must consider the dynamics of non-condensed magnons similarly to the two-component model of superfluid helium.

Finally, taking into account the value of the condensate mobility $\eta$ obtained from the fit of the experimental data and the known effective mass $m$, we calculate the effective time, describing the relaxation of the linear momentum of the condensate $\approx 10$ μs. Surprisingly, the obtained time is much longer than the lifetime of the condensate $\tau = 130$ ns. Although this phenomenon is not yet understood, one can speculate that it can be an indication of the onset of the magnetic superfluidity.

In conclusion, our experimental results together with a simple model provide a direct evidence that the magnon BEC in YIG films is intrinsically stable with respect to the collapse in the real space. The origin of this stability is the effective repulsive magnon–magnon interaction having the magneto-dipolar nature. Additionally, our findings demonstrate an important role of dissipative flows in magnon condensates, caused by the momentum dissipation due to the interaction of

the condensate with non-condensed magnons. We firmly believe, that our results will stimulate further experimental and theoretical exploration of the temporal and spatial dynamics of magnon BEC.

## Methods

**Test system**. The magnon BEC was studied in a 5.1 μm-thick YIG film epitaxially grown on a Gadolinium Gallium Garnet (GGG) substrate. A strong microwave field necessary for magnon injection was created by using a dielectric microwave resonator with the resonant frequency $f_p = 9.055$ GHz, fed by a microwave source. The device was placed into the uniform static magnetic field $H_0 = 0.5$–1.5 kOe. Since the results obtained at different fields were qualitatively very similar, in the main text, we only discuss the data obtained at $H_0 = 0.64$ kOe. An additional spatially inhomogeneous magnetic field was created by using a Au control line carrying dc electric current (width 10 μm, thickness 400 nm, and length 8 mm). The line was fabricated on a sapphire substrate. The substrate was placed between the resonator and the YIG sample in such way that the conductor was directly attached to the surface of the YIG film (see Fig. 1a, b).

**Micro-focus and time-resolved BLS measurements**. All the measurements were performed at room temperature. We focused the single-frequency probing laser light with the wavelength of 532 nm onto the surface of the YIG film through a transparent substrate and analysed the light inelastically scattered from magnons. The measured signal—the BLS intensity—is directly proportional to the magnon density $n$. By varying the lateral position of the laser spot, we recorded spatial profiles of $n$. To analyse the spatio-temporal dynamics of the condensate free from the influence of pumping, the pumping field was applied in pulses with the duration of 1 μs and the period of 10 μs and the BLS signal was recorded as a function of the time delay with respect to falling edge of the pulse.

**Dipolar field due to local increase of the magnon density**. The field was calculated both analytically and by using micromagnetic simulations. The spatial distribution of the magnon density $\Delta n(z)$ was exemplary taken in the form of a Gaussian distribution. Analytically, the field was calculated using the Green-function approach[32], with the Green function representing the magnetic field of infinitely thin layer (cross-section of the film) of magnetic charges at a given $z$. The resulting magnetic field was then found as a convolution of the above Green function with the magnetic charge density distribution, which is proportional to the spatial derivative of the magnetization or the magnon density $dM/dz \propto dn/dz$. The micromagnetic simulations were performed by using the software package Mumax3.[33] The computational domain with dimensions of $5 \times 20 \times 100$ μm$^3$ was discretized into $1 \times 0.1 \times 0.1$ μm$^3$ cells. The standard value for the exchange stiffness of $3.7 \times 10^{-12}$ J/m was used. Periodic boundary conditions were applied at all lateral boundaries. The results obtained from the analytical and numerical calculations agree with a precision of 3% (Fig. 4b).

## Data availability

The data that support the findings of this study are available from the corresponding author upon reasonable request.

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

## Acknowledgements

This work was supported in part by the Deutsche Forschungsgemeinschaft (Project No. 416727653). The theoretical work was supported by the University of Cologne Center of Excellence QM2 and by William R. Thurman'58 Chair in Physics, Texas A&M University.

## Author contributions

I.V.B. and V.E.D. performed measurements and data analysis, I.V.B, G.L., T.N., and V.L.P. developed the theoretical model, B.D. performed micromagnetic simulations, S.O.D. formulated the idea of the experiment and managed the project. All authors co-wrote the manuscript.

## Competing interests

The authors declare no competing interests.
