## [Peer Review File · Nature Communications]

Reviewers' comments:

Reviewer #1 (Remarks to the Author):

This is a well written paper that describes experiments that investigate the relaxation of a high density of magnons in inhomogeneous magnetic fields. The authors aim to understand the stability of Bose-Einstein condensates of magnons, such as whether there are indeed the repulsive interactions between magnons needed to prevent a collapse of a magnon BEC.

In the experiment a small decrease in magnetic field causes magnon accumulation and the authors study the density profile and the magnon relaxation rate in this potential. Based on a saturation of the density for an increasing confining potential (more negative magnetic field perturbation) and the profile of the density as the driving rf field is turned off, the authors infer a repulsive interaction between magnons.

A saturation of the magnon density can reflect that the maximum density of magnons set by the film's magnetization (magnetic moment/volume). I recommend the authors estimate their maximum magnon density relative to maximum possible density.

There is fundamental theory that shows that magnon-magnon interactions are *attractive* in materials with perpendicular magnetic anisotropy (e.g. Ivanov and Kosevich, JETP Lett. 24, 501 (1976)). There are also experiments reporting the observation of magnetic solitons created by spin-transfer torques that are associated with this attractive interaction between magnons in perpendicular magnetic thin films, e.g., D. Backes et al. PRL 115, 127205 (2015) & S. Chung et al., PRL 120, 217204 (2018). These paper could be cited along with the present references 21-23 on magnetic solitons. Thus it appears that the magnetic anisotropy, in-plane, e.g. in YIG or out of plane plane a critical role in whether there are attractive or repulse magnon-magnon interactions. I recommend that the authors comment on the importance of the magnetic anisotropy in their experiment and its analysis.

In sum I find the paper interesting and the analysis convincing. I think the paper is suitable for publication in Nature Communications.

Reviewer #2 (Remarks to the Author):

In this manuscript, the authors have experimentally shown that the Bose-Einstein condensed magnons in yttrium-iron garnet (YIG) films are repulsively interacting, stabilizing the magnon condensate. In the experiment, the condensate is confined by a spatially varying magnetic field. The dependence of the condensate density profile on the potential depth and the number of condensed atoms (after turning the microwave pumping off) indicates that the inter-magnon interaction is repulsive, differently from the theoretical prediction. The obtained density profile shows good agreement with the mean-field calculation. The authors attribute the origin of the repulsive interaction to the dipolar field caused by the non-uniform distribution of the condensate.

The paper is well-organized and clearly written. Though I am working in the field of AMO physics, I could smoothly read the paper. The experimental data is convincing, supported by theoretical calculation and interpretation. I think this paper gives a great impact on the field of magnon condensation as it gives the solution for the issue of the long-time stability of magnon Bose-Einstein condensates. I'm willing to recommend the publication in Nature communications.

I have several minor questions and comment, which I would like the authors to address before publication:

1) I could not well understand how to create the potential by the magnetic field gradient. Because the minimum of the magnon dispersion depends on the strength of the magnetic field, the magnetic field gradient is expected to cause the spatially dependent potential. However, the wavelength corresponding to the magnon minimum (which is also dependent on the magnetic field) is in the same order of the potential width. I guess the authors used a kind of local-density-approximation, but I wonder how it is valid.

2) The authors argued that the repulsive interaction is caused by the dipolar field created by the non-uniform distribution of the condensate. Then, naively thinking, the interaction is expected to be anisotropic. Is it possible to see whether the interaction along the x-axis is attractive or repulsive, by observing the density distribution along the x-axis? The interaction is also expected to be sensitive to the relative direction of the confinement and the polarization. I am curious about how the result changes if, for example, the uniform field H_0 is applied in the x direction. (I think, if the potential is determined by the Zeeman energy, the field gradient induced by the control line works as a confinement along the z direction.)

3) Related to the above question, I'd like to understand what happens in the absence of the potential. Suppose that a spatially uniform condensate first appears. Then the attractive interaction induces the density modulation. The density wave develops such that the resulting repulsive interaction via the dipolar field stabilizes the condensate. If this picture is true, is it possible to observe a stable density wave structure in the experiment?

Reviewer #3 (Remarks to the Author):

In the submitted manuscript, "Direct evidence of spatial stability of Bose-Einstein condensate of magnons", I.V Borisenko et al. study the room-temperature Bose-Einstein condensation of magnons in thin YIG films. This very interesting problem has a long history and one of the issues, still questioning the possibility of BEC of magnons in thin films, is the condensate stability, related to the actual sign of the magnon's scattering amplitude (i.e., to the sign of the effective inter-magnon interaction). The manuscript reports experimental results presenting new evidences for spatial stability of the condensate. In addition, since the established theories predict unstable condensate for experimental conditions as in the manuscript (due to the attractive effective interaction between magnons), the authors present theoretical model supporting their experimental observations and propose a mechanism, potentially responsible for changing the sign of the scattering amplitude from

negative to positive.

In my opinion, the clear experimental evidence of the condensate stability, supported by correct theoretical model, would definitely deserve publication in the Nature Communications (to re-iterate, this is a long standing and very interesting and important problem). However, before making the final conclusion, I have to ask a few questions.

(1) In the description of Fig.2b the authors state that the density of magnons “tends to saturate at large well depths”. Actually, this cannot be clearly concluded from Fig. 2b – the slope changes but saturation cannot be really seen. If authors have more data supporting their claim, it would be very helpful to present it.

(2) In the Discussion the authors mention that the lifetime of the condensate and magnon life-time due to the spin-lattice relaxation are “far below $1 \mu s$ ”. Simultaneously, the authors use the Fig.3, to prove the repulsive nature of inter-magnon interaction in the condensate. Note, however, that at least the 660 ns curve in Fig. 3(a) can be considered as being very close to “far below $1 \mu s$ ” life-time of magnons and condensate. This can make conclusions, based on Fig.3, somehow questionable and requires the corresponding explanation in the text.

(3) In the theoretical description the authors use the 1-d Gross-Pitaevskii equation with a few parameters to calculate the condensate density in the case where the density itself is non-uniform. According to the manuscript, the non-uniform distribution of density induces the dipolar demagnetizing field which results in the actual repulsive inter-magnon interaction required to prove the stability of condensate. Below are a few theoretical questions:

(a) The values of two parameters, η and g , are defined in the text. However, the value of τ , Eq.6, is not given. This parameter, accounting for the processes of repopulation and depletion of condensate, should be specified as well.

(b) In the Fig. 2a the normalized theoretical density curve is presented together with the experimental one for the case of potential well ($\Delta H = -10$ Oe). Can the experimental curve for potential hill (+10 Oe) be theoretically reproduced with the same values of the above 3 parameters η , g , and τ ? If it is possible, the corresponding curve should be presented as it adds to demonstration of the model’s reliability. If it is not possible, it has to be explained.

(c) Actually, the magnon scattering amplitude is magnetic field dependent while the manuscript assumes it to be constant. However, at magnetic fields as in the manuscript the amplitude’s field-dependence is very slow. Thus, the coupling g can indeed be approximated by a constant if one does not expect any strong change of total magnetic field. As it has been stated by the authors, the previous theoretical considerations were limited to the case of spatially uniform condensate. The model, presented in the manuscript, takes into account the non-uniformity of condensate and calculates the demagnetizing field, induced by the above spatial non-uniformity. The calculated and measured demagnetizing field is reported to be about 2 Oe which looks absolutely reasonable. Since the experiment itself is performed in the magnetic field $H_0 = 640$ Oe, the 2 Oe change of total field on top of the constant external 640 Oe makes the model assumption of field-independent coupling g

acceptable. However, according to the authors the non-uniformity of the condensate is what differs their model from all previous theoretical descriptions, predicting attractive inter-magnon interaction and unstable condensate for experimental conditions as in the manuscript. Thus, it looks rather strange that the extra 2 Oe of the dipolar demagnetizing field on top of the external 640 Oe can change the sign of the scattering amplitude in comparison with what previous models predict, as authors propose (having in mind that in the manuscript the coupling g is assumed to be field-independent). If the authors have an explanation on how it is possible, it should be presented in the main text since otherwise the proposed mechanism of changing the sign of scattering amplitude due to the induced (very small in comparison with the external field) demagnetizing field does not look convincing.

Reply to Reviewer #1

This is a well written paper that describes experiments that investigate the relaxation of a high density of magnons in inhomogeneous magnetic fields. The authors aim to understand the stability of Bose-Einstein condensates of magnons, such as whether there are indeed the repulsive interactions between magnons needed to prevent a collapse of a magnon BEC. In the experiment a small decrease in magnetic field causes magnon accumulation and the authors study the density profile and the magnon relaxation rate in this potential. Based on a saturation of the density for an increasing confining potential (more negative magnetic field perturbation) and the profile of the density as the driving rf field is turned off, the authors infer a repulsive interaction between magnons.

We thank the Reviewer for the positive evaluation of our work and for the constructive comments, which we thoroughly addressed in the revised manuscript, as described below.

A saturation of the magnon density can reflect that the maximum density of magnons set by the film's magnetization (magnetic moment/volume). I recommend the authors estimate their maximum magnon density relative to maximum possible density.

After having read the comments by all Reviewers, we understood that the term “saturation”, used in the original manuscript to characterize the experimental findings in Fig. 2b is not exact. In fact, keeping in mind the log scale of the figure, the dependence is far from saturation at the values of ΔH used in the experiment. It is more exact to say, that one observes a crossover from the fast exponential growth to the slow one. We have reformulated the corresponding sentence on page 5. Since the magnon density is far from the true saturation, its magnitude is far below the maximum possible density of magnons. Following the Reviewer's inquiry, we have estimated the maximum possible density corresponding to a 100%-reduction of the magnetization, 10^{22} cm^{-3} . In the revised manuscript, we have indicated this value on page 11, and mentioned that the experimental density is by three orders of the magnitude smaller than the maximum density.

There is fundamental theory that shows that magnon-magnon interactions are *attractive* in materials with perpendicular magnetic anisotropy (e.g. Ivanov and Kosevich, JETP Lett. 24, 501 (1976)). There are also experiments reporting the observation of magnetic solitons created by spin-transfer torques that are associated with this attractive interaction between magnons in perpendicular magnetic thin films, e.g., D. Backes et al. PRL 115, 127205 (2015) & S. Chung et al., PRL 120, 217204 (2018). These paper could be cited along with the present references 21-23 on magnetic solitons. Thus it appears that the magnetic anisotropy, in-plane, e.g. in YIG or out of plane plane a critical role in whether there are attractive or repulse magnon-magnon interactions. I recommend that the authors comment on the importance of the magnetic anisotropy in their experiment and its analysis.

We thank the Reviewer for pointing out this issue. We fully agree with Reviewer that the anisotropy results in dependence of the energy of magnons on their density and determines the type of magnon-magnon interactions. In the papers mentioned by the Reviewer, the

attractive interaction is caused by the dominating uniaxial anisotropy. In our case, it is due to the dipolar shape anisotropy of the film magnetized parallel to its plane.

To comply with the Reviewer's comment, in the revised manuscript, we have cited the mentioned papers (Refs. 18, 25, 26) and indicated on page 2 that, in our case, the origin of the attractive magnon interactions is the dipolar shape anisotropy of the in-plane magnetized film.

In sum I find the paper interesting and the analysis convincing. I think the paper is suitable for publication in Nature Communications.

We thank the Reviewer for the recommendation to publish our paper in Nature Communications.

Reply to Reviewer #2

In this manuscript, the authors have experimentally shown that the Bose-Einstein condensed magnons in yttrium-iron garnet (YIG) films are repulsively interacting, stabilizing the magnon condensate. In the experiment, the condensate is confined by a spatially varying magnetic field. The dependence of the condensate density profile on the potential depth and the number of condensed atoms (after turning the microwave pumping off) indicates that the inter-magnon interaction is repulsive, differently from the theoretical prediction. The obtained density profile shows good agreement with the mean-field calculation. The authors attribute the origin of the repulsive interaction to the dipolar field caused by the non-uniform distribution of the condensate.

The paper is well-organized and clearly written. Though I am working in the field of AMO physics, I could smoothly read the paper. The experimental data is convincing, supported by theoretical calculation and interpretation. I think this paper gives a great impact on the field of magnon condensation as it gives the solution for the issue of the long-time stability of magnon Bose-Einstein condensates. I'm willing to recommend the publication in Nature communications.

I have several minor questions and comment, which I would like the authors to address before publication:

We thank the Reviewer for the positive evaluation of our work and the recommendation to publish it in Nature Communications. As described in detail below, in the revised manuscript, we have taken into account all the Reviewer's questions and comments.

1) I could not well understand how to create the potential by the magnetic field gradient. Because the minimum of the magnon dispersion depends on the strength of the magnetic field, the magnetic field gradient is expected to cause the spatially dependent potential. However, the wavelength corresponding to the magnon minimum (which is also dependent on the magnetic field) is in the same order of the potential width. I guess the authors used a kind of local-density-approximation, but I wonder how it is valid.

We thank the Reviewer for pointing out this issue. Now we see that we have forgotten to indicate the wavelength of the condensate in the original manuscript. In fact, this wavelength is about $1\ \mu\text{m}$ and it does not change noticeably within the used range of magnetic fields. This wavelength is significantly smaller than the characteristic widths of the non-uniform potential of about $20\ \mu\text{m}$, which proves the validity of the used local-density approximation. To address the Reviewer's comment, we have indicated the wavelength of the condensate on page 4 of the revised manuscript.

2) The authors argued that the repulsive interaction is caused by the dipolar field created by the non-uniform distribution of the condensate. Then, naively thinking, the interaction is expected to be anisotropic. Is it possible to see whether the interaction along the x-axis is attractive or repulsive, by observing the density distribution along the x-axis? The interaction is also expected to be sensitive to the relative direction of the confinement and the polarization. I am curious about how the result changes if, for example, the uniform field H_0 is applied in the x direction. (I think, if the potential is determined by the Zeeman energy, the field gradient induced by the control line works as a confinement along the z direction.)

We would like to emphasize that, according to our model, the interaction should be repulsive for both z - and x -direction, once a z -inhomogeneity of the condensate density is formed. Indeed, any local increase of the magnon density along the x -direction leads to an additional local increase of dipolar field oriented in the z -direction, which locally increases the energy per magnon resulting in magnon repulsion. In agreement with this model, in the experiment, the condensate density was found to be uniform along the control line.

We agree with the Reviewer that it is interesting to study magnon interactions as a function of the relative orientation of the confinement direction and the static magnetic field. We would like to note, however, that it is very challenging from the experimental point of view. Indeed, an additional confinement field, which is not parallel to the static field, has a weaker influence on the total field: for the parallel case $H_{\text{tot}} = H_0 + \Delta H$, whereas, e.g., for the perpendicular case $H_{\text{tot}} = \sqrt{H_0^2 + \Delta H^2}$. Keeping in mind that $\Delta H \ll H$, the variation of the magnon energy, and, correspondingly, the confinement effects are much weaker in the latter case. Therefore, in this first work, we restrict ourselves by the one-dimensional confinement along the applied field.

3) Related to the above question, I'd like to understand what happens in the absence of the potential. Suppose that a spatially uniform condensate first appears. Then the attractive interaction induces the density modulation. The density wave develops such that the resulting repulsive interaction via the dipolar field stabilizes the condensate. If this picture is true, is it possible to observe a stable density wave structure in the experiment?

We fully agree with the Reviewer. This is exactly the picture one expects in the case of an unconfined condensate. However, since the initial attractive interaction is very weak for magnons in the vicinity of the lowest-energy state, the characteristic spatial scale of such a density-wave structure is estimated to be of the order of few millimeters. Due to the experimental limitations, it is very difficult to create magnon BECs with both spatial

dimensions exceeding few hundreds of micrometers. Therefore, by now, this phenomenon could not be observed experimentally.

Reply to Reviewer #3

In the submitted manuscript, “Direct evidence of spatial stability of Bose-Einstein condensate of magnons”, I.V Borisenko et al. study the room-temperature Bose-Einstein condensation of magnons in thin YIG films. This very interesting problem has a long history and one of the issues, still questioning the possibility of BEC of magnons in thin films, is the condensate stability, related to the actual sign of the magnon’s scattering amplitude (i.e., to the sign of the effective inter-magnon interaction). The manuscript reports experimental results presenting new evidences for spatial stability of the condensate. In addition, since the established theories predict unstable condensate for experimental conditions as in the manuscript (due to the attractive effective interaction between magnons), the authors present theoretical model supporting their experimental observations and propose a mechanism, potentially responsible for changing the sign of the scattering amplitude from negative to positive.

In my opinion, the clear experimental evidence of the condensate stability, supported by correct theoretical model, would definitely deserve publication in the Nature Communications (to re-iterate, this is a long standing and very interesting and important problem). However, before making the final conclusion, I have to ask a few questions.

We thank the Reviewer for the positive evaluation of our work and emphasizing the importance of the studied problem. Below we answer the Reviewer’s questions and describe how they have been taken into account in the revised manuscript. We hope that Reviewer will find our answers convincing and will recommend publication of the revised manuscript in Nature Communications.

(1) In the description of Fig.2b the authors state that the density of magnons “tends to saturate at large well depths”. Actually, this cannot be clearly concluded from Fig. 2b – the slope changes but saturation cannot be really seen. If authors have more data supporting their claim, it would be very helpful to present it.

We agree with the Reviewer that the term “saturation” does not correctly describe the observed behaviors. In fact, the true saturation of the dependence in Fig. 2b is not necessary to support our claim that the expected attractive interaction is not active in the studied system. If the interactions were attractive, one would expect the confinement of the condensate in the well to stimulate accumulation of magnons in the real space followed by a collapse. This should result in a very fast increase of the condensate density with the increase of the well depth. In contrast, the experimental data of Fig. 2b show that the variation of the density with the increase of the magnitude of $\Delta_H < 0$ becomes slower, which clearly proves the absence of magnon attraction. To make this clearer, we have reformulated the corresponding sentence on page 5 of the revised manuscript.

(2) In the Discussion the authors mention that the lifetime of the condensate and magnon life-time due to the spin-lattice relaxation are “far below 1 μ s”. Simultaneously, the authors use the Fig.3, to prove the repulsive nature of inter-magnon interaction in the condensate. Note, however, that at least the 660 ns curve in Fig. 3(a) can be considered as being very close to “far below 1 μ s” life-time of magnons and condensate. This can make conclusions, based on Fig.3, somehow questionable and requires the corresponding explanation in the text.

We thank the Reviewer for drawing our attention to this issue. In fact, the lifetime of the condensate $\tau=130$ ns is defined as the time, during which the condensate density decreases by a factor of e (see also our reply to the question 3(a) below). Therefore, it is not surprising that one can still observe condensate at delays of 660 ns. Thanks to the high sensitivity of the BLS apparatus, we are able to follow the evolution of the condensate density for temporal intervals significantly exceeding τ (see Fig. 3b). To make this clear, we have added in the text of the revised manuscript (page 8) the definition of τ . To avoid any misunderstanding we explicitly stated that the parameter τ in Eq. 6 was not a fit parameter and explained how it was measured separately (p. 9). We have also reformulated the sentence on page 11, mentioned by the Reviewer.

(3) In the theoretical description the authors use the 1-d Gross-Pitaevskii equation with a few parameters to calculate the condensate density in the case where the density itself is non-uniform. According to the manuscript, the non-uniform distribution of density induces the dipolar demagnetizing field which results in the actual repulsive inter-magnon interaction required to prove the stability of condensate. Below are a few theoretical questions:

(a) The values of two parameters, η and g , are defined in the text. However, the value of τ , Eq.6, is not given. This parameter, accounting for the processes of repopulation and depletion of condensate, should be specified as well.

We thank the Reviewer for pointing out this omission. In the revised manuscript (page 9), we specify the value $\tau=130$ ns and indicate that this value was separately determined: the temporal dependence of the condensate density of the homogeneous condensates was studied after abrupt switching off the pumping. Correspondingly, we used the value $\tau=130$ ns as a fixed parameter in all our simulations based on Eq. 6. Note here, that this time nicely agrees with the data shown in Fig. 3b for $z=5 \mu\text{m}$. The relaxation time measured at the center of the well, 180 ns is longer than 130 ns, whereas the time measured at $z=10 \mu\text{m}$, 120 ns is shorter. This difference is another experimental evidence that a redistribution of the condensate between different spatial points takes place in our system. In the revised manuscript, we additionally indicated the determined values of the effective relaxation time close to the curves in Fig. 3b.

(b) In the Fig. 2a the normalized theoretical density curve is presented together with the experimental one for the case of potential well ($\Delta H = - 10$ Oe). Can the experimental curve for potential hill (+ 10 Oe) be theoretically reproduced with the same values of the above 3 parameters η , g , and τ ? If it is possible, the corresponding curve should be presented as it adds to demonstration of the model’s reliability. If it is not possible, it has to be explained.

Following the Reviewer's request, we have added in Fig. 2a the theoretical curve for the case of a potential hill and discussed it on pages 5 and 10 of the revised manuscript. We emphasize that τ is not a fitting parameter. We use $\tau=130$ ns, as determined from experiments with the uniform condensate. For calculation of both curves, the same value of g is used. Our simulations show that the width of the profile is mainly determined by the mobility of the condensate η . Therefore, to fit the experimental curve for the case of the hill (which is apparently broader than that for the case of the well), we need to increase the value of η . This is not surprising keeping in mind that the magnon densities differ between the two cases by a factor of 30. Note that, for example, in classical gases, where the mobility is determined by inter-atomic scattering, it is inversely proportional to the gas density. Our system is more complex: as indicated in the paper, we believe that the momentum relaxation of the condensate dynamics is mainly determined by the scattering of the condensate from non-condensed magnons. If the density of non-condensed magnons were the same for the two cases, the mobility would also be the same. However, since the field inhomogeneity also influences the density of non-condensed magnons, the mobility itself is a complex function of the magnetic field and of the magnon densities. Since the scattering processes are not yet well understood, instead of writing a complex system of differential equations rigorously considering all possible contributions, we use a relatively simple Eq. 6, aimed to describe gross features of the phenomenon.

(c) Actually, the magnon scattering amplitude is magnetic field dependent while the manuscript assumes it to be constant. However, at magnetic fields as in the manuscript the amplitude's field-dependence is very slow. Thus, the coupling g can indeed be approximated by a constant if one does not expect any strong change of total magnetic field. As it has been stated by the authors, the previous theoretical considerations were limited to the case of spatially uniform condensate.

The model, presented in the manuscript, takes into account the non-uniformity of condensate and calculates the demagnetizing field, induced by the above spatial non-uniformity. The calculated and measured demagnetizing field is reported to be about 2 Oe which looks absolutely reasonable. Since the experiment itself is performed in the magnetic field $H_0 = 640$ Oe, the 2 Oe change of total field on top of the constant external 640 Oe makes the model assumption of field-independent coupling g acceptable.

However, according to the authors the non-uniformity of the condensate is what differs their model from all previous theoretical descriptions, predicting attractive inter-magnon interaction and unstable condensate for experimental conditions as in the manuscript. Thus, it looks rather strange that the extra 2 Oe of the dipolar demagnetizing field on top of the external 640 Oe can change the sign of the scattering amplitude in comparison with what previous models predict, as authors propose (having in mind that in the manuscript the coupling g is assumed to be field-independent). If the authors have an explanation on how it is possible, it should be presented in the main text since otherwise the proposed mechanism of changing the sign of scattering amplitude due to the induced (very small in comparison with the external field) demagnetizing field does not look convincing.

We would like to emphasize that, in the case of spatially uniform density of magnons, the scattering amplitude for magnons with frequencies close to the lowest-energy state is very small (see Ref. 33). The corresponding frequency shift is only -0.14 MHz (compare with +6 MHz obtained in this work). The reason for this is that the frequency of the lowest state is

very close to the Larmor frequency $f_0 = \gamma H_0$. The latter depends on the static magnetic field H_0 , but is INDEPENDENT of the magnon DENSITY. Since magnon scattering amplitude is determined by the dependence of the energy of magnons on their density, for magnons with the frequency f_0 , the scattering amplitude is ZERO at ANY value of the static field H_0 .

In contrast, non-uniform spatial distribution of magnons results in an appearance of the dipolar field, which linearly DEPENDS on the magnon density resulting in a density-dependent magnon energy. In other words, the effect of the non-uniformity-induced demagnetizing dipolar field on the magnon scattering amplitude is superior to that of the static uniform magnetic field H_0 .

To comply with the Reviewer's comment, we have extended the discussion on pages 10-11 of the revised manuscript.

REVIEWERS' COMMENTS:

Reviewer #1 (Remarks to the Author):

I believe that the authors have adequately addressed the comments and questions in my initial report.

Reviewer #2 (Remarks to the Author):

The authors have sincerely responded to my questions and comments. Now I understand the points I was concerning, and I am willing to recommend publication in Nature communications.

Reviewer #3 (Remarks to the Author):

The authors addressed all the questions that I asked. I accept all these answers (and corresponding modifications of the manuscript) except for the last theoretical one. I understand that their theoretical treatment, Eq.6, is too approximate and therefore it is not easy to reproduce both experimental curves in Fig.2 just by changing the sign of magnetic field, as it should be in the case of accurate theoretical description. Nevertheless, as a "qualitative demonstration" it can do the job. However, I am not convinced by the author's explanation about changing the sign of scattering amplitudes. First of all, internal demagnetizing field adds up to the external one and magnon scattering amplitudes depend on total field (as well as on film thickness). Second, quantum treatment of the full problem's Hamiltonian takes into account demagnetizing effects even in the uniform case since both exchange and dipole-dipole interaction terms contribute to both scattering amplitudes and energy. Third, at the film thickness and magnetic field as in the author's experiment the field-dependence of relevant amplitudes is very slow and amplitudes are not very small. All this can be found in theoretical works that authors quote in the manuscript.

However, I did not do calculations with non-uniform spatial distribution and cannot guaranty that there are no extra effects leading to changing the sign of amplitudes. Simultaneously, experimental evidences and other explanations look rather reasonable. This is why I would recommend the manuscript for publication with one condition. The authors should add their theoretical explanation of the sign of amplitudes into the text, at least in the same way as they did it in the Reply to me. It will explain why authors believe that small demagnetizing field can change the overall sign of field-dependent scattering amplitude. Also, in this way it will be easier for other groups to verify the picture presented in the manuscript.

To conclude, I am not convinced by the author's explanation about changing sign of scattering amplitudes solely due to demagnetizing field. However, I would recommend publication of the manuscript in the Nature Communications if authors add their explanation on how the sign changes (at least in the same way as they explained it to me in the Reply) into the text. The latter will present the author's view and allow other groups to verify the picture.

Reviewer #1 (Remarks to the Author):

I believe that the authors have adequately addressed the comments and questions in my initial report.

We thank the reviewer for his/her evaluation of our paper.

Reviewer #2 (Remarks to the Author):

The authors have sincerely responded to my questions and comments. Now I understand the points I was concerning, and I am willing to recommend publication in Nature communications.

We thank the reviewer for his/her evaluation of our paper and suggestion to publish our paper.

Reviewer #3 (Remarks to the Author):

The authors addressed all the questions that I asked. I accept all these answers (and corresponding modifications of the manuscript) except for the last theoretical one. I understand that their theoretical treatment, Eq.6, is too approximate and therefore it is not easy to reproduce both experimental curves in Fig.2 just by changing the sign of magnetic field, as it should be in the case of accurate theoretical description. Nevertheless, as a "qualitative demonstration" it can do the job. However, I am not convinced by the author's explanation about changing the sign of scattering amplitudes. First of all, internal demagnetizing field adds up to the external one and magnon scattering amplitudes depend

on total field (as well as on film thickness). Second, quantum treatment of the full problem's Hamiltonian takes into account demagnetizing effects even in the uniform case since both exchange and dipole-dipole interaction terms contribute to both scattering amplitudes and energy. Third, at the film thickness and magnetic field as in the author's experiment the field-dependence of relevant amplitudes is very slow and amplitudes are not very small. All this can be found in theoretical works that authors quote in the manuscript.

However, I did not do calculations with non-uniform spatial distribution and cannot guaranty that there are no extra effects leading to changing the sign of amplitudes. Simultaneously, experimental evidences and other explanations look rather reasonable. This is why I would recommend the manuscript for publication with one condition. The authors should add their theoretical explanation of the sign of amplitudes into the text, at least in the same way as they did it in the Reply to me. It will explain why authors believe that small demagnetizing field can change the overall sign of field-dependent scattering amplitude. Also, in this way it will be easier for other groups to verify the picture presented in the manuscript.

To conclude, I am not convinced by the author's explanation about changing sign of scattering amplitudes solely due to demagnetizing field. However, I would recommend publication of the manuscript in the Nature Communications if authors add their explanation on how the sign changes (at least in the same way as they explained it to me in the Reply) into the text. The latter will present the author's view and allow other groups to verify the picture.

As suggested by the Reviewer, we added in the text the argumentation on the role of a small dipole field on the scattering amplitude, which has been already presented in our previous Reply.